# The Circulating Nucleic Acid Characteristics of Non-Metastatic Soft Tissue Sarcoma Patients

**DOI:** 10.3390/ijms21124483

**Published:** 2020-06-24

**Authors:** Nicholas Eastley, Aurore Sommer, Barbara Ottolini, Rita Neumann, Jin-Li Luo, Robert K. Hastings, Thomas McCulloch, Claire P. Esler, Jacqueline A. Shaw, Robert U. Ashford, Nicola J. Royle

**Affiliations:** 1Trauma and Orthopaedics, University Hospitals of Leicester NHS Trust, Leicester LE1 5WW, UK; Robert.Ashford@uhl-tr.nhs.uk; 2Department of Genetics and Genome Biology, University of Leicester, Leicester LE1 7RH, UK; as1000@leicester.ac.uk (A.S.); barbara.ottolini@diadx.com (B.O.); rn20@leicester.ac.uk (R.N.); jl19@leicester.ac.uk (J.-L.L.); rkh7@leicester.ac.uk (R.K.H.); js39@leicester.ac.uk (JAS); njr@leicester.ac.uk (N.J.R.); 3Nottingham University Hospitals NHS Trust, Nottingham NG5 1PB, UK; Tom.McCulloch@nuh.nhs.uk (T.M.); Claire.Esler@nuh.nhs.uk (C.P.E.)

**Keywords:** Genetics, cell free DNA, soft tissue sarcoma

## Abstract

Soft tissue sarcomas (STS) are rare, malignant tumours with a generally poor prognosis. Our aim was to explore the potential of cell free DNA (cfDNA) and circulating tumour DNA (ctDNA) analysis to track non-metastatic STS patients undergoing attempted curative treatment. The analysed cohort (*n* = 29) contained multiple STS subtypes including myxofibrosarcomas, undifferentiated pleomorphic sarcomas, leiomyosarcomas, and dedifferentiated liposarcomas amongst others. Perioperative cfDNA levels trended towards being elevated in patients (*p* = 0.07), although did not correlate with tumour size, grade, recurrence or subtype, suggesting a limited diagnostic or prognostic role. To characterise ctDNA, an amplicon panel covering three genes commonly mutated in STSs was first trialled on serial plasma collected from nine patients throughout follow-up. This approach only identified ctDNA in 2.5% (one in 40) of the analysed samples. Next custom-designed droplet digital PCR assays and Ion AmpliSeq™ panels were developed to track single nucleotide variants identified in patients’ STSs by whole exome sequencing (1–6 per patient). These approaches identified ctDNA in 17% of patients. Although ctDNA was identified before radiologically detectable recurrence in two cases, the absence of demonstrable ctDNA in 83% of cases highlights the need for much work before circulating nucleic acids can become a useful means to track STS patients.

## 1. Introduction

Soft tissue sarcomas (STSs) are a diverse group of malignant tumours that originate from mesenchymal tissues. Although their incidence is rising [1] STSs remain relatively rare, with only 13,100 new cases diagnosed annually in the European Union, equating to just 1% of all new adult cancer diagnoses [2,3]. STSs are classified into two groups based on their genomic characteristics [4]. The first group includes around 15 different STS subtypes and 20% of cases overall. This group of tumours are characterised by their near-diploid genomes and subtype-specific oncogene activating translocations or ring chromosomes. The second group comprises the majority of STSs, which are characterised by complex, unstable genomes, and a wide range of abnormalities including chromosome copy-number changes (polyploidy or aneuploidy), unbalanced translocations, amplifications, deletions and chromothripsis.

The curative treatment of STSs centres on their surgical resection, commonly combined with (neo)adjuvant radiotherapy. Following this approach, a significant proportion of high-grade STSs will recur either locally (17%) or with metastatic disease (24%) [5,6]. To date, no recognised circulating biomarkers of STS tissue are available for clinical use. An unfortunate consequence of this is that, despite surveillance, this STS recurrence is often extensive when diagnosed, leaving patients palliative options alone.

Circulating cell-free DNA (cfDNA) is defined as extracellular nucleic acids that circulate freely in the blood stream. In patients with cancer a proportion of cfDNA is shed directly from tumoural tissue and is termed circulating tumour DNA (ctDNA). In many cancers total cfDNA levels [7,8,9,10] and ctDNA characteristics [11,12,13] have been shown to correlate with tumour behaviour and patient outcome. The potential usefulness of ctDNA in patient management has been explored in some STS patients with metastatic disease, notably leiomyosarcomas [14,15] among others [16,17,18,19]. However, to date only one study has investigated ctDNA characteristics in non-metastatic patients with myxoid or well-differentiated/dedifferentiated liposarcomas [20]. To address this paucity of data, we performed a prospective longitudinal study investigating the cfDNA/ctDNA characteristics of a cohort of non-metastatic STS patients undergoing attempted curative treatment, using three different approaches to quantify ctDNA.

## 2. Results

The aim of this study was to characterise cfDNA and ctDNA in patients diagnosed with primary (non-metastatic) STS at the time of surgery to determine whether they can be used to monitor a change to disease status during the follow-up period. Among the 29 patients recruited to the study, 10 were diagnosed with myxofibrosarcoma (MFS), seven with undifferentiated pleomorphic sarcoma (UPS); five with leiomyosarcoma (LMS), two with de-differentiated liposarcoma (DDLS), and the remainder singletons of other subtypes (Table 1 and Appendix A). The majority of the patients had a large tumour burden at diagnosis (Table 1, average volume 544 cm^3^). Considering this we initially investigated total cfDNA levels as a potential biomarker for monitoring STS patients with non-metastatic disease at diagnosis.

### 2.1. Comparison of Peri- and Post-Operative cfDNA Levels Among STS Patients

Intra-operative plasma samples were collected from 25 patients (Appendix A). The mean cfDNA concentration in these patients was 12.3 ng/mL plasma (±2.1SEM, median 9.3, range 2.3–43.6, Appendix A). No correlation was seen between intra-operative cfDNA concentration and tumour size (*R*^2^ = 0.19), Trojani grade (*p* = 0.9, ANOVA) or STS subtype (*p* = 0.96, ANOVA) (Appendix A). No difference was seen in the mean intra-operative cfDNA level of those patients that developed recurrence and those that remained disease free during follow up (10.5 ng/mL vs. 12.8 ng/mL, *p* = 0.67, unpaired *t*-test, Appendix A).

Post-operative plasma samples were collected from 26 patients (Appendix A). The mean first post-operative cfDNA level for these patients was 8.8 ng/mL plasma (±1.2 SEM, median 7.6, range 2–30). Matched intra- and post-operative plasma samples were available for 22 patients, but no significant drop in cfDNA level was seen following surgery in this group (*p* = 0.51, paired *t*-test; Figure 1). There was no significant difference between the mean post-operative cfDNA level of those patients that suffered recurrence, and those that remained disease free during follow-up (Appendix A, *p* = 0.85, unpaired *t*-test). There was also no significant difference in the actual difference between these groups’ intra- and post-operative cfDNA levels (Appendix A, *p* = 0.85, unpaired *t*-test). Similarly, there was no significant difference when these comparisons were made between those patients that had wide or marginal resections (Appendix A, *p* = 0.97/0.61, unpaired *t*-test).

Finally, cfDNA levels at first post-operative appointment and at the point of disease recurrence was available for eight out of nine patients that recurred during follow up. Although mean cfDNA levels were 2.2 ng/mL higher at recurrence, this difference was not significant (Figure 1b, *p* = 0.41, paired *t*-test).

### 2.2. Targeted NGS of Patients’ Tumour and ctDNA

We initially sought to develop a ctDNA assay that would be widely applicable for monitoring STS patients, by screening the three most commonly mutated genes in STSs (*TP53*, *RB1* and *ATRX*; www.cbioportal.org). To characterise ctDNA we developed an Ion Torrent AmpliSeq™ panel custom designed to track the most common somatic SNVs in these genes (‘Sarcoma V2’, Appendix A). This panel was used on plasma samples collected from nine patients, six of whom had matched STS tissue DNA also available for analysis (Table 1). Only two of the six STSs investigated showed evidence of SNVs in the targeted genes. These somatic SNVs were seen at a frequency of 1% (patient 17, *ATRX;* D2106G) and 18% (patient 32, *TP53;* C135F) (Table 2.) Neither of these SNVs was identified in matched plasma samples. However, somatic SNVs were detected at a low level in two other patients (patient 21 and 23) (Table 1 and Table 2, Figure 2).

#### 2.2.1. Patient 21

Patient 21 (F/77 yrs) underwent neoadjuvant radiotherapy and a wide surgical resection for a grade 2 myxoid liposarcoma. During her 83-week follow up, she remained disease-free, and provided 4 plasma samples for analysis 1, 10, 28 and 40 weeks post-operatively. Single low frequency circulating somatic SNVs were identified in plasma collected intra-operatively (*RB1*; E884E, 0.9%) and 10 weeks post-operatively (*RB1*; I388T, 0.5%) but at no later timepoints. The significance of the *RB1*; E884E SNV is unclear, but is unlikely to have contributed to sarcoma formation in this patient. No matched tumour tissue was available for analysis.

#### 2.2.2. Patient 23

Patient 23 (M/53 yrs) had a grade 2 undifferentiated pleomorphic sarcoma also treated with neoadjuvant radiotherapy and a wide surgical resection. During his 70-week follow-up, he remained disease free, and provided 4 plasma samples for analysis 2, 21, 35 and 49 weeks post-operatively. Two low-frequency circulating somatic SNVs were detected in plasma collected 21 weeks post-operatively (*TP53*; V274V, 0.6%, *TP53*; intronic variant, 0.7%) but not subsequently. Neither variants were identified in patient 23’s STS tissue, and neither are expected to have contributed to STS formation.

### 2.3. Patient-Specific ddPCR Analysis of Tumour and ctDNA

Given the genetic heterogeneity of STSs and the limited ctDNA detection achieved using our custom designed three-gene ‘Sarcoma V2’ ampliseq panel, we next adopted a personalised approach to detect ctDNA. For this, patients’ tumours were initially analysed using comparative WES to identify somatic SNVs present. Next, ddPCR assays designed to identify a selection of these variants in the circulation were used to analyse matched patient plasma samples (Appendix A). In total serial plasma samples collected from eight patients were analysed in this way, with one or two SNV specific ddPCR assays used in each case (Table 1 and Table 3). Despite adopting this new approach, intra- or post-operative ctDNA was only identified in two patients (patient 6 and 22, both diagnosed with UPS).

#### 2.3.1. Patient 6

Patient 6 (M/54 years) had a grade 2 UPS managed with neoadjuvant radiotherapy and a wide surgical resection. Following surgery, he was followed up for 33 weeks before he developed metastatic (pulmonary) recurrence. During this time, he provided plasma samples for analysis intra-operatively, and four, 11, 29 and 33 weeks post-operatively. Two somatic SNVs were tracked in patient 6’s plasma-*TP53*:R306 * and *BRIP1*:P971A. Circulating *TP53*:R306 * was identified in plasma samples collected 11 (VAF 0.42%, 254 copies/mL) and 29 (VAF 1.75%, 3580 copies/mL) weeks post-operatively (Figure 3 and Appendix A). No evidence of circulating *BRIP*1:P971A was identified in any of the plasma samples collected.

#### 2.3.2. Patient 22

Patient 22 (F/64 years) had a grade 2 UPS treated with neoadjuvant chemotherapy, a wide surgical resection and adjuvant radiotherapy. Follow-up lasted 54 weeks until a diagnosis of metastatic (pulmonary) recurrence was made. During this time plasma was collected intra-operatively and 5, 23, 37 and 53 weeks post-operatively. Two SNVs were tracked in patient 22’s plasma, *EPHB6*:G397R and *DACH1*:G594D. Circulating *EPHB6*:G397R was identified intra-operatively (VAF 0.85%) and in samples collected 37 (VAF 1.6%, 7919 copies/mL) and 53 weeks post operatively (VAF 2.1%) (Figure 3 and Appendix A). No evidence of circulating *DACH1*:G594D was identified in any of the plasma samples collected. In summary, using truly personalised, sensitive ddPCR assays we detected ctDNA in 25% (2/8) of patients analysed. However, the characteristics of this ctDNA correlated with disease progression in just one patient (patient 22).

### 2.4. Patient Specific tNGS Analysis of Intra Operative Plasma Samples 

Although our personalised approach to the characterisation of ctDNA using ddPCR assays showed some promise, the proportion of patients with measureable levels of ctDNA was low. Therefore, we next elected to seek ctDNA in plasma collected intra-operatively, based on a hypothesis that it may be more abundant at the point of maximal disease burden. To enable us to track multiple patient-specific variants in each case, we developed another custom designed IonTorrent panel, comprising three Ampliseq primer pools (‘Sarcoma V345’, Appendix A).

#### 2.4.1. Tumour and Plasma Analysis

‘Sarcoma V345’ tNGS panel was used to analyse STS tissue and intra-operative plasma collected from 10 patients (Table 1). Thirty selected SNVs (range 1−6/patient) were identified in tumour samples at similar frequencies to those detected by WES analysis (Appendix A). The ‘Sarcoma V345’ panel was used to profile the same SNVs in the patients’ matched intra-operative plasma samples (Appendix A). This analysis identified ctDNA in just one patient (patient 43).

#### 2.4.2. Patient 43

Patient 43 (77 yr/M) had a large (2160 cm^3^) grade 2 myxofibrosarcoma treated with neoadjuvant radiotherapy followed by wide local excision. Four circulating variants were identified in their intraoperative plasma at frequencies of 0.65–1.19%. These involved *ABCC5* (*R729W*, 1.19%), *TRIO* (H950Y, 0.65%), *PLAG1* (D380Y, 0.93%), and *HSPA9* (G430D, 0.83%) (Table 4).

## 3. Discussion

Soft tissue sarcoma patient survival has been static for most tumour subtypes for the last 25 years. Several barriers must be addressed to overcome this, one of which is a lack of STS biomarkers to monitor patients during their treatment. To investigate the potential role of circulating nucleic acids for this purpose, our group has previously characterised cfDNA/ctDNA in metastatic STS patients [16]. Here, we present our analysis of these characteristics in a cohort of non-metastatic STS patients.

The mean intra-operative cfDNA level was 12.3 ng/mL (*n* = 25), although wide variation was seen between individuals (range 2.3−43.6 ng/mL). This is higher than previously published values in healthy adults [16], although not significantly so (*p* = 0.07, Appendix A). We are only aware of one published cfDNA level for a non-metastatic STS patient, which was 110 ng/mL in a patient with a large (878 cm^3^) tumour one day prior to resection [18]. Data is similarly scarce for other malignancies, although levels of 8.0−344 ng/mL (non-small cell lung cancer [21]), 0.5–235 ng/mL (breast cancer [22]), 30.1 ng/mL (colorectal cancer [23]) and 59 ng/mL (pancreatic cancer [24]) suggest that non-metastatic STS patients may have lower cfDNA levels than other cancer patients. We found no correlation between intra-operative cfDNA levels and STS size or grade. This may be explained by (1) the absence of high levels of ctDNA, (2) variation in the ctDNA characteristics of individual STS subtypes, or (3) the presence of ctDNA shed disproportionately to STS size or grade, potentially due to varying contributions of apoptotic, necrotic or cfDNA secreting cells [25].

Patient cfDNA levels at their first post-operative follow-up appointments (*n* = 26) were also higher than those seen in healthy individuals [16], but not significantly so (*p* = 0.07, Appendix A). Cell-free DNA levels only dropped post-operatively by 1.3 ng/mL in the cohort (*n* = 22). Considering cfDNA’s rapid clearance from the circulation [26,27], this may be explained by an absence of notable intra-operative ctDNA, despite large tumour volumes (average 544 cm^3^). Alternatively, the long interval between many of the patients’ surgeries and first post-operative appointments (mean 28 days) may have concealed more significant cfDNA drops in the immediate post-operative period. In Patient 31, the cfDNA level rose noticeably following surgery from 5.8 to 30 ng/mL. In the absence of an obvious clinical explanation, this is likely due to sample contamination by lysed white blood cell DNA. Although excluding 31’s data increased the mean post-operative drop in cfDNA to 2.5 ng/mL, this fall remains smaller than that reported in other malignancies (35 ng/mL in colorectal cancer for example [28]).

No difference was seen in the intra-operative, post-operative or peri-operative drop in cfDNA levels of those patients that suffered disease recurrence, and those that remained disease-free during follow-up. In the metastatic recurrent patients analysed (*n* = 8) cfDNA levels rose just 2.2 ng/mL between patients’ first post-operative appointments, and the point disease recurrence was identified. Excluding patient 31’s data, increased this difference to 4.4 ng/mL (*p* = 0.017, paired *t*-test). Overall however, considering the natural physiological fluctuance seen in cfDNA levels [29], our data suggest that cfDNA levels alone have little potential as a clinically useful biomarker of STS recurrence.

To characterise ctDNA we initially designed an IonTorrent AmpliSeq™ panel (Sarcoma V2) targeting the three most commonly mutated genes in STSs and tested intra-operative and multiple post-operative plasma samples from nine patients with non-metastatic STSs. Using this panel, we detected four circulating somatic variants in plasma collected from two out of nine patients (patients 21 and 23; 2 SNVs per patient). STS tissue was available for patient 23, but neither circulating SNV identified in this case was identified in the matched tumour tissue (Table 2). Moreover, three of the four detected SNVs were silent or intronic, and therefore of unknown significance with respect to STS development or progression.

Given the genetic heterogeneity among STS subtypes we next developed multiple patient-specific ddPCR assays to profile circulating SNVs, with the aim of increasing the likelihood of detecting ctDNA. Using this individualised approach, ctDNA was identified in 2/8 patients analysed (25%) each with just one of the two SNVs selected to track (patient 6 and 22, Table 3). Intriguingly, both these patients, initially diagnosed with non-metastatic grade 2 UPS, showed evidence of ctDNA when radiologically ‘free of disease’ (Figure 3). In one case (patient 22), this was over 31 weeks before the detection of disease recurrence, highlighting the kind of scenario in which ctDNA may act as a marker of micrometastatic STS disease. Despite this, our inability to detect ctDNA using a sensitive and personalised approach in 75% of patients is disappointing, especially given the success with which ddPCR has been used to characterise ctDNA in other malignancies [30].

Based on the hypothesis that ctDNA may be more abundant in patients at the point of maximal disease burden, we next focused on ctDNA detection in plasma collected from patients during primary tumour resection. For this we developed a second IonTorrent AmpliSeq panel (‘Sarcoma V345’) custom-designed to target multiple SNVs identified at a high frequency in each patient’s STSs by WES analysis. This approach was even less successful at identifying ctDNA than using ddPCR, but allowed us to detect several low circulating tumoural variants, albeit in just 1/10 patients analysed (10%) (Table 4 and Appendix A). Notably, in this one patient, four of five selected SNVs were detected at similarly low levels (0.65–1.19%), providing reassurance that ctDNA was present in the analysed intra-operative plasma sample.

Among the eight patients analysed using the longitudinal personalised ddPCR approach, five patients showed disease progression during the follow-up period (Table 1). In only one of five patients (20%) did the detection and abundance of ctDNA potentially facilitating the prediction of recurrence (patient 22, UPS). Profiling ctDNA was also unhelpful in predicting survival, with intra- or post-operative ctDNA not identified in either of the patients that died during follow up (one UPS /one MFS). The low predictive value of ctDNA analysis in our study may be a reflection of the small number of patients that showed disease progression in our cohort, and the varied histological subtypes analysed (Table 1). Other more focused studies of individual STS subtypes have used selected/subtype specific mutations to track ctDNA with disease progression in 34–50% of patients analysed [15,20]. In one of these studies focusing on LMS patients, ctDNA was also identified in a much higher proportion of patients with metastatic disease than stable or low disease burden disease (11/16, 69%vs. 0/16, 0%). Our failure to identify ctDNA in all three of the LMS patients we analysed (two with stable disease, one with progressive disease) is consistent with this.

We have used three different experimental approaches to detect and monitor ctDNA across a diverse range of STS subtypes, which has reduced our capacity to investigate differences between various subtypes. Nevertheless, we note that ctDNA (validated by prior detection of the SNVs in the matched STS tissue) was detected in one patient with a myxofibrosarcoma (MFS) and two patients with undifferentiated pleomorphic sarcoma (UPS) in our cohort. Circulating tumoural DNA has not been reported previously in patients with an MFS, but it has been detected in up to 30% (2/6) of patients with UPS [19]. Further work will be needed to explore whether patients with UPS are more likely to shed ctDNA than patients with other less common subtypes.

In our study, patients in which ctDNA was detected using a personalised approach (three out 18, 17%), circulating variants remained at a low frequency. This may reflect a true absence of ctDNA, or alternatively the limited release of circulating nucleic acids from the subclones containing the SNVs selected to profile during our analysis. Regardless of which explanation is accurate, this low frequency and low incidence of ctDNA suggests that ctDNA analysis in patients with non-metastatic disease is unlikely to offer any means to diagnose STS recurrence.

The work presented here is the first sizable longitudinal study of the circulating nucleic acid characteristics of a cohort of non-metastatic STS patients. We show that targeted sequencing of genes commonly mutated in STSs has little value as a screening tool or longitudinal characterisation of ctDNA in patients with non-metastatic primary STS (consistent with a previous study on metastatic STS [19]). We have also shown that using two alternative patient-specific approaches to detect circulating SNVs already identified at a high frequency in analysed patients’ tumours also appears to have limited value for patients with non-metastatic STS. This conclusion is based primarily on our inability to identify ctDNA in 83% (15 out of 18) of patients analysed, and so to discriminate between STS patients and healthy individuals. This failure may be explained in two ways. Firstly, the majority of patients analysed may have indeed had no ctDNA, confirming that nucleic acids are not shed into the circulation consistently by STSs. Secondly, our experimental approach may have not been specific enough to identify any ctDNA present, although this seems unlikely given the variety of patient-specific, experimental approaches used. Despite the low frequency and low level of ctDNA detection in STS patients it remains possible that ctDNA monitoring could be useful in a small subset of patients but moving forward a key challenge will be to determine which patients might benefit.

## 4. Methods

### 4.1. Ethics and Registration

The project was approved by the National Health Service National Research Ethics Service (NRES) Committee (REC reference: 14/NE/1192, IRAS project ID: 141820, 22/10/14) and was conducted in accordance with the Declaration of Helsinki. It was also publically registered on www.ClinicalTrials.gov (Identifier: NCT02547376).

### 4.2. Patient Enrolment

A cohort of 29 patients with biopsy proven non-metastatic STSs scheduled to undergo attempted curative surgical resections were enrolled for analysis (Table 1). Every patient provided informed written consent prior to providing samples. Appendix A shows the full inclusion and exclusion criteria. The patients identified as ‘white British’ except for patient 9 (Indian) and patient 30 (‘white, other’).

### 4.3. Patient Assessment

At each follow-up appointment patients underwent a chest radiograph looking for pulmonary metastases (the most common site for STS metastases) and were examined for local and regional recurrence. All radiographs were reviewed by a consultant musculoskeletal radiologist with a specialist interest in soft tissue tumours.

### 4.4. Tissue Collection

Multiple tumour samples were collected from each patient’s STS immediately following resection. Samples were snap frozen in liquid nitrogen and stored at –80 °C. Prior to any analysis each sample was macro- and microscopically assessed to ensure they were representative of viable STS tissue.

### 4.5. Blood Collection

One 20 mL whole blood sample was collected in a standard potassium-EDTA tube from each patient during surgery to resect their STS, prior to tumour removal. Serial 20 mL whole blood samples were subsequently collected from patients at each of their routine post-operative clinical follow-up appointments. This generally consisted of one appointment 2 weeks after surgery, followed by 3 monthly appointments. Each whole blood sample was processed to isolate plasma and buffy coat as previously described [31].

### 4.6. DNA Extraction and Quantification

Tumour genomic DNA was isolated from STS tissue with a verified tumour cell content, by Proteinase K digestion followed by phenol-chloroform extraction. Total cfDNA was extracted from 3 mL plasma using the QIAamp Circulating Nucleic Acid kit (Qiagen, Hilden, Germany). Buffy Coat (BC) DNA was extracted using the QIAamp DNA Blood Mini Kit (Qiagen, Hilden, Germany). Tumour DNA and BC DNA yields were determined using the Qubit^®^ dsDNA HS (High Sensitivity) Assay Kit and a Qubit^®^ 2.0 Fluorometer (Thermo Fisher Scientific, Massachusetts, MA, USA). Circulating free DNA yields were quantified using real-time qPCR [32].

### 4.7. IonTorrent SNV Panel Design

Two targeted next generation sequencing (tNGS) panels were custom designed to analyse patient samples (‘Sarcoma V2’ and ‘Sarcoma V345’). ‘Sarcoma V2’ was designed to cover the commonest non-synonymous exonic or splice site single nucleotide variants (SNVs) in *TP53*, *ATRX*, and *RB1*, the 3 most commonly mutated genes in STSs according to cBioportal data (Appendix A). This panel had 45 amplicons averaging 84 bp in length (range 68–96) of which 6 required exclusion from our analysis for poor amplification efficiency (defined as a mean read depth of <1000.) ‘Sarcoma V345’ comprised three pools of primers (Appendix A) designed to analyse patient-specific somatic non-synonymous exonic or splice site SNVs identified by comparative tumour vs normal whole exome sequencing (WES, Appendix A). SNVs previously associated with sarcomas or other cancers were preferentially selected to target in this way (range 1–6 SNVs per patient). Altogether the Sarcoma V345 primer pools covered 46 amplicons averaging 47 bp in length (range 31–59), although 16 of these required exclusion due to low amplification efficiency or failure to meet other quality control measures (Appendix A). The tNGS panels were validated by comparing the frequency of the SNVs derived from WES analysis of patient tumour tissue with their frequency obtained using the AmpliSeq panel.

### 4.8. Semiconductor tNGS and Somatic Variant Calling

tNGS was performed using an Ion Torrent Personal Genome Machine (PGM) sequencer (Life Technologies, California, United States). DNA libraries were prepared using the Ion AmpliSeq™ Library Kit v2.0 (ThermoFisher, California, CA, USA) using 10ng of template DNA. Pooled barcoded libraries were prepared for sequencing using the Ion PGM™ Hi-Q™ View OT2 Kit (Thermo Fisher Scientific Waltham, MA, USA), and template sequenced on the Ion PGM™ System using the Ion PGM™ Hi-Q™ View Sequencing Kit using Ion 314 and 316™ chips. Somatic variants were identified using IonTorrent Variant Caller software and verified by manual review of Bam files using the Integrated Genomics Viewer (IGV) package (v2.3.25) available from The Broad Institute (http://software.broadinstitute.org/software/igv/). The presence of circulating somatic variants at a frequency of >0.5% in patient plasma was provisionally deemed evidence of ctDNA. Matched patient BC DNA was sequenced to gauge background sequencing noise, and confirm the somatic nature of any circulating variants identified.

### 4.9. Droplet Digital PCR Assay Development for Tumour Derived SNVs

In addition to the two tNGS panels, patient-specific droplet digital PCR (ddPCR) assays were also developed to track ctDNA in plasma samples collected at surgery and throughout follow up. Each selected SNV identified at a variant frequency of >20% by WES analysis was also subsequently verified in STS DNA using standard PCR and Sanger sequencing. Taqman hydrolysis probe ddPCR assays (Eurogentec, Liège, Belgium) were developed and optimised in house for the selected SNVs (Appendix A). Droplet digital PCR primer specificity was confirmed using standard PCR and Evagreen ddPCR using the QX200™ Droplet Digital™ PCR System (Appendix A). In addition, a commercial validated ddPCR assay was used to detect and quantify circulating *TP53*; *R306** in patient 006 (Bio-Rad Laboratories, California, CA, USA).

### 4.10. SNV ddPCR Reaction Conditions

SNV ddPCR was performed using the QX200™ Droplet Digital™ PCR System as per the manufacturer’s instructions (Appendix A). Prior to plasma analysis the optimal cycling conditions and sensitivities of each ddPCR assay was established using patient tumour and BC DNA as template.

## Figures and Tables

**Figure 1 ijms-21-04483-f001:**
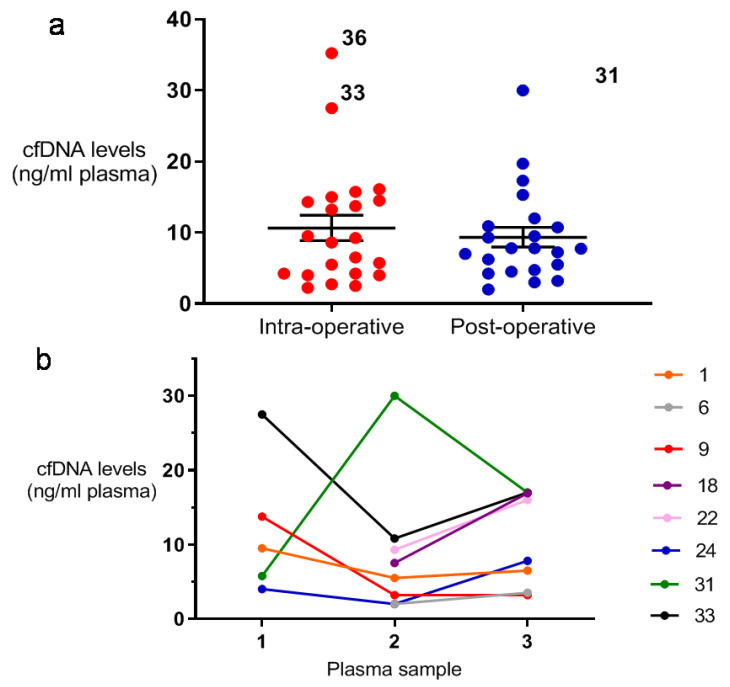
Analysis of cfDNA level in STS patients. (**a**) Peri-operative cfDNA levels in STS patients. The mean intra-operative cfDNA levels in patients with matched intra- and post-operative cfDNA levels available for analysis was 10.7 ng/mL (±1.8 SEM, median 8.9, range 2.3−35.3) compared with 9.4 ng/mL (±1.4 SEM, median 7.8, range 2−30) post-operatively (*p* = 0.51, paired *t* test). Participant numbers for the outliers are shown. (**b**) Longitudinal trends in cfDNA levels in recurrent STS patients. Intra-operative cfDNA levels (plasma sample 1), cfDNA levels at patients’ first post-operative follow up appointment (plasma sample 2) and cfDNA levels at the point their recurrence was diagnosed (plasma sample 3) are shown. Patients’ mean cfDNA levels rose by 2.2 ng/mL between their first post-operative appointment and the point when recurrence was diagnosed (mean of 8.8 ± 3.2 SEM and 11.0 ± 2.2 SEM respectively). Intra-operative cfDNA levels were not available for patients 6, 18 or 22.

**Figure 2 ijms-21-04483-f002:**
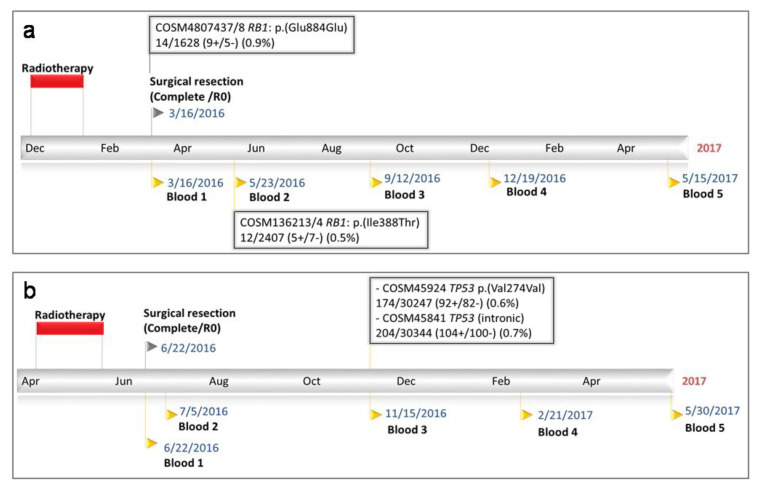
Circulating SNVs detected using tNGS (‘Sarcoma V2’) in STS patients. (**a**,**b**) show data for patient 21 and 23 respectively.

**Figure 3 ijms-21-04483-f003:**
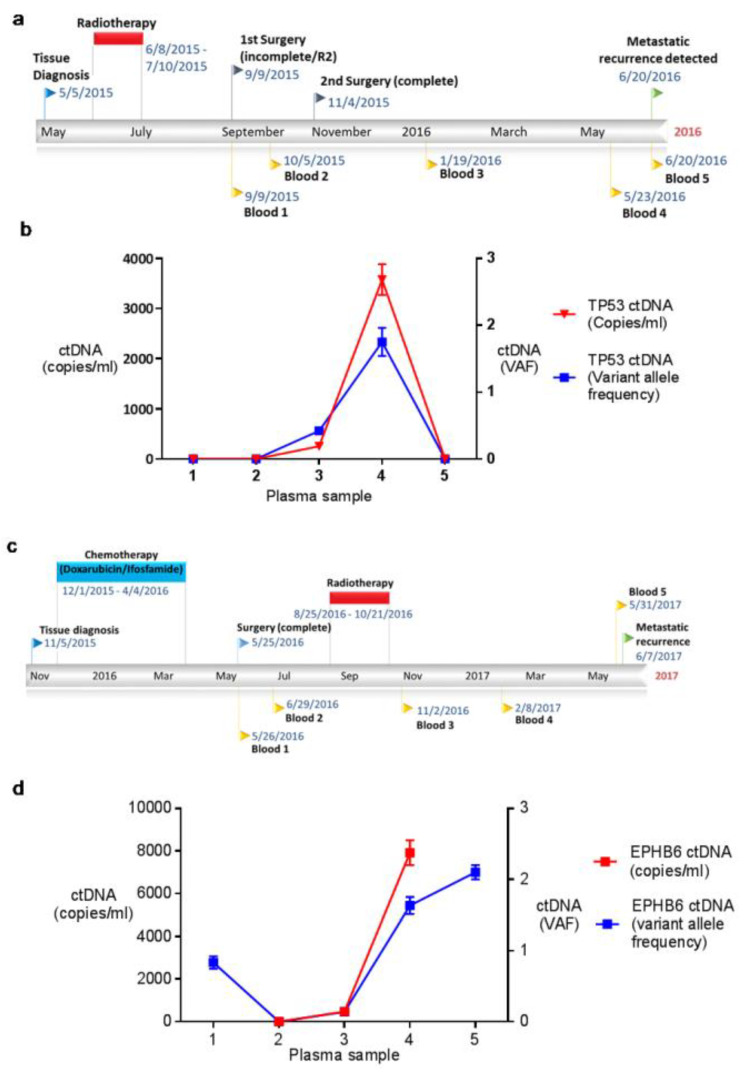
Management, outcome and ctDNA characteristics of patients 6 and 22. (**a**) The timeline of patient 6’s progress from diagnosis throughout treatment and follow-up until recurrence including dates of plasma samples collected. (**b**) shows the levels of circulating *TP53*; R306* identified in each plasma sample from patient 6. Circulating *TP53*:R306 * was identified in samples 3 (VAF 0.42%, 254 copies/mL), and 4 (VAF 1.75% (SEM 0.15), 3580 copies/mL (SEM 307)). (**c**) The timeline of patient 22’s progress from the point of diagnosis throughout treatment and follow-up. (**d**) shows the levels of circulating plasma *EPHB6*:G397R in each plasma sample from patient 22. Circulating *EPHB6*:G397R was identified at a VAF of >0.5% intra-operatively (VAF 0.85% (SEM 0.15) and in plasma samples 4 (VAF 1.6% (SEM 0.12), 7919 copies/mL (SEM 582)) and 5 (VAF 2.1% (SEM 0.1)). It was not possible to calculate the absolute number of copies of *EPHB6*:G397R intra-operatively or in plasma sample 5 as total cfDNA levels were unavailable for these samples. In Figure 3b,d the right axis represents variant allele circulating fractional abundance and the left axis the variant allele concentration in copies/mL. Error bars showing standard error of the mean (SEM) are shown where biological replicates were possible.

**Table 1 ijms-21-04483-t001:** Summary of patient cohort, sarcoma features, treatment and ctDNA analysis.

Pt	Age (yrs)/Gender	STS Subtype. (+Trojani Tumour Grade)	Tumour Volume (cm^3^)	Radiotherapy/Chemotherapy (Nil/Neo/Adj)	STS Recurrence	Oncology Outcome	WES	No. Samples Screened for ctDNA: Intra-op (Post-op)	Mode of Analysis. (No. SNVs Screened).	ctDNA Detected. (No. SNVs Detected).	ctDNA Predictive
1	76.1/M	MFS (3)	663	Rad (Neo)	Metastatic	AWD	Yes	1 (5).	tNGS-V2.	/	/
3	63.1/M	ExMC (unknown)	588	Rad (Adj)	No	NED	Yes	1 (4).	ddPCR (1)	No (0)	/
6	55.3/M	UPS (2)	8	Rad (Neo)	Metastatic	AWD	Yes	1 (4).	ddPCR (2)	Yes (1)	No
9	62.3/F	LMS (3)	Unknown	Rad (Neo)	Metastatic	AWD	Yes	1 (1).	ddPCR (2)	No (0)	No
10	59.7/F	SS (2)	9	Nil	No	NED	No	1 (6).	tNGS-V2	/	/
17	27.6/M	MFS (1)	539	Nil	No	NED	No	1 (4).	tNGS-V2	/	/
18	80.0/F	HS (unknown)	Unknown	Nil	Metastatic	AWD	Yes	1 (2).	ddPCR (1)	No (0)	No
21	76.5/F	MLS (2)	198	Rad (Neo)	No	NED	No	1 (4).	tNGS-V2	Yes (1) Uncertain ^a^	/
22	65.4/F	UPS (3)	364	Chemo (Neo) & Rad(Adj)	Metastatic	AWD	Yes	1 (4).	ddPCR (2)	Yes (1)	Yes
23	53.2/M	UPS (2)	117	Rad (Neo)	No	NED	No	1 (4).	tNGS-V2	Uncertain ^b^	/
24	68.9/M	MFS (2)	144	Rad (Adj)	Metastatic	DOD	Yes	1 (2).	ddPCR (1)	No	No
25	36.7/F	UPS (3)	630	Nil	unknown.	Lost to FU	Yes	1 (2).	ddPCR (2)	No	/
26	62.8/M	DDLS (2)	759	Rad (Neo)	No	NED	Yes	1 (3).	ddPCR (1)	No	/
27	67.0/F	UPS (2)	129	Rad (Neo)	No	NED	Yes	/	/	/	/
28	70.6/F	MFS (2)	113	Nil	No	NED	No	1 (2).	tNGS-V2.	/	/
29	74.0/M	LMS (3)	525	Rad (Neo)	No	NED	Yes	/	/	/	/
30	22.2/M	ES (3)	151	Chemo (Neo) & Rad(Adj)	unknown.	Lost to FU	Yes	/	/	/	/
31	45.8/M	UPS (3)	2947	Rad (Neo)	Metastatic	DOD	Yes	1 (3)	tNGS-V2 & V345(3)	No	/
32	64.0/F	MFS (3)	4	Nil	No	NED	No	1 (1)	tNGS-V2 (1)	No	/
33	79.7 /M	LMS (3)	3289	Rad (Neo/Adj)	Metastatic	AWD	Yes	1 (2)	tNGS-V2	/	/
34	69.0 /M	UPS (2)	27	Nil	Local.	NED	Yes.	1	tNGS-V345 (3)	No	/
35	87.2/F	MFS (3)	38	Nil	No	NED	Yes	1	tNGS-V345 (2)	No	/
36	74.2/M	DDLS (2)	576	Rad (Neo).	No	NED	Yes	1	tNGS-V345 (4)	No	/
37	74.4/F	MFS (2)	9	Nil	No	NED	Yes	1	tNGS-V345 (1)	No	/
38	48.7/F	LMS (2)	61	Rad (Adj)	No	NED	Yes	1	tNGS-V345 (3)	No	/
40	70.3/M	MFS (3)	68	Rad (Adj)	No	NED	Yes	1	tNGS-V345 (5)	No	/
41	81.2/F	MFS (3)	70	Rad (Adj)	No	NED	Yes	1	tNGS-V345 (3)	No	/
43	77.0/M	MFS (2)	2160	Rad (Neo)	No	NED	Yes	1	tNGS-V345 (5)	Yes (4)	/
44	74.0/M	LMS (3)	506	Rad (Neo)	No	NED	Yes	1	tNGS-V345 (1)	No	/

M-Male; F-Female; Myxofibrosarcoma (MFS); Extraskeletal Myxoid Chondrosarcoma (ExMC); Undifferentiated Pleomorphic Sarcoma (UPS); Leiomyosarcoma (LMS); Synovial Sarcoma (SS); Haemangiosarcoma (HS); Myxoid Liposarcoma (MLS); Dedifferentiated Liposarcoma (DDLS); Soft Tissue Ewing’s Sarcoma (ES); Adj-Adjuvant; Neo-Neoadjuvant; AWD-alive with disease; DOD-died from disease; NED-no evidence of disease; FU-follow up. ^a^ Two *RB1* SNVs detected, one non-synonymous SNV was detect in inter-operative sample and the second SNV of uncertain significance (silent) was detected in the 10 week post-operation sample. ^b^ Two *TP53* SNVs of uncertain significance (silent or intronic) were detected only in the 21week post-operation.

**Table 2 ijms-21-04483-t002:** SNVs identified in *TP53*, *RB1* and *ATRX* in STS or cfDNA from patients.

								STS Tissue DNA.			cfDNA DNA.				
Pt	Chr.	Location of SNV	Gene	Coding Strand	Base Chang	Cosmic ID	Predicted Effect	Depth (Reads)	Variant Reads	%	Depth (Reads)	Variant Reads (%)	Total cfDNA (ng/mL)		Plasma Sample. (Weeks Post-op)
17	X	76814213	*ATRX*	-	T > C.	4971451/2	p.(Asp2106Gly)	5477	23+/28−	1%	Not detected.				
21	13	49050968	*RB1*	+	A > G	4807437/8	p.(Glu884Glu)	Not available			1628	9+/5− (0.9%)	13.3.	IO	
	13	48947576	*RB1*	+	T > C	136213/4	p.(Ile388Thr)				2407	5+/7− (0.5%)	5.8	PO (10)	
23	17	7577116	*TP53*	-	T > C	1386598/45924	p.(Val274Val)	Not detected			30247	92+/82− (0.6%)	5.0	PO (21)	
	17	7578346	*TP53*	-	G > A	45841	Intronic	Not detected			30344	104+/100− (0.7%)			
32	17	7578526	*TP53*	-	G > T	303849-52	p.(Cys135Phe)	25409	2003+/2534−	18%	Not detected				

Data is shown from the analysis of patient samples using the tNGS sarcoma V2 AmpliSeq™ panel. IO–intra-operative; PO–post-operative. The SNVs: *RB1* A > G p.(Gly884Glu) in patient 21 and both *TP53* SNVs in patient 23 are of unknown significance with regard to STS development. X: Chromosome X.

**Table 3 ijms-21-04483-t003:** SNVs targeted in STS patient plasma using ddPCR. ^a^ Mutation frequencies from WES analysis. ^b^ The Sorting Intolerant From Tolerant software package (SIFT) was used to predict the effect of each SNV, D–damaging, T–tolerated, - no prediction offered.

Patient Number	Gene	SNV Position (Chr:Loci)	Coding Strand (+/−)	Base Change	Predicted Effect	Mutation Frequency in STS ^a^	SIFT Prediction ^b^	SNV Detected in Matched Plasma
3	*VWDE*	7:12384078	-	T>C	Cys1302Arg	42%	D	N
6	*TP53*	17:7577022	-	C>T	Arg306Ter	56%	-	Y
	*BRIP1*	17:59761496	-	C>G	Pro971Ala	20%	T	N
9	*PTCH1*	9: 98239884	-	C>A	Ala332Glu	23%	D	N
	*LPP*	3:188327063	+	C>A	Pro182Thr	46%	D	N
18	*FLT4*	5: 180046092	-	G>A	Val927Met	18%	D	N
22	*DACH1*	13: 72053389	-	A>C	Glu594Asp	21%	-	N
	*EPHB6*	7: 142563798	+	G>A	Gly397Arg	44%	-	Y
24	*MMS22L*	6: 97634424	-	C>T	Gln728Ter	25%	-	N
25	*ITIH2*	10: 7769692	+	C>T	Arg394Trp	37%	D	N
	*KDM5B*	1: 202777369	-	C>T	Pro22Leu	88%	D	N
26	*PTPRB*	12:70970320	-	C>T	Thr677Ile	73%	T	N

**Table 4 ijms-21-04483-t004:** Tumour and plasma variants identified in STS patient analysed using targeted NGS (‘Sarcoma V345′).

							STS tissue DNA			cfDNA		
Pt	Chr	Location of SNV	Gene	Coding Strand	Base Change	Predicted Effect	Depth (Reads)	Variant reads	%	Depth (Reads)	Variant Reads (%)	Total cfDNA (ng/mL)
43	3	183681223	*ABCC5*	-	G > A	p.(R729W)	2889	629+/1053−	58.22%	3939	11+/36− (1.19%)	43.6
	5	14367062	*TRIO*	+	C > T	p.(H950Y)	17322	4071+/2949−	40.53%	5670	23+/14− (0.65%)	
	8	57078921	*PLAG1*	-	C > A	p.(D380Y) p.(D462Y)	5230	864+/1488−	44.21%	2259	12+/9− (0.93%)	
	5	137895674	*HSPA9*	-	C > T	p.(G430D)	25278	6605+/5019−	45.98%	5936	23+/26− (0.83%)	

Data is shown from the analysis of patients’ intra-operative plasma samples using the sarcoma V345 AmpliSeq™ panel.

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
