# Peer review of "The Circulating Nucleic Acid Characteristics of Non-Metastatic Soft Tissue Sarcoma Patients"

_ijms, 2020, doi:10.3390/ijms21124483_

Round 1

Reviewer 1 Report

The manuscript entitled “The circulating nucleic acid characteristics of Non-Metastatic Soft Tissue Sarcoma patients” by Nicholas Eastley, et al. describes the potential role of cell-free DNA (cfDNA) and circulating tumor DNA (ctDNA) as biomarkers for monitoring STS patient outcome. The authors provide a prospective longitudinal study in a cohort of non-metastatic STS patients using different approaches to quantify ctDNA. However, the approaches used to analyze the circulating nucleic acid appear to have limited value and applicability in clinical routine, due to the low frequency and low levels of ctDNA detection.

The paper is well written and the search strategy and the study selection process are well designed. The manuscript has a potential to be accepted, but some major and minor points have to be clarified or fixed before.

Major revision:

Although STSs are relatively rare, the cohort of STS patients included in this study is limited. I suggest to the authors to include more samples that can ameliorate the statistical analysis and the conclusions. Moreover, including more patients can improve also the correlation between intra-operative cfDNA levels respect to healthy adults.

The experimental approaches conducted in this study include different histotypes in the same analyses. The authors can explain the results also for each histotype in the Results and Discussion section.

In the Results section, the Patient-specific ddPCR paragraph appears confused. The somatic SNV identified by comparative WES should be listed (Line 135) to explain the specific SNV tracked in patients.

Table 3 is not included in the manuscript.

Minor revision:

In lines 75, 79, 212, in Supplementary Fig 1 and Supplementary Fig 4, the letter P must be lowercase.

Line 261, the word UPS should be correct.

In line 201, “Supplementary 1” should be changed in “supplementary Fig 1”.

In line 73, the author should include the statistical correlation.

In line 204, the n. 4 after the refs 22 should be deleted as well as n. 19 in line 281.

In line 146, BRIP1, P971A should be corrected.

Author Response

We also thank reviewer 1 for their positive comments and for giving us the opportunity to address the points they have raised.

Reviewer 1 major revision i):

‘Although STSs are relatively rare, the cohort of STS patients included in this study is limited. I suggest to the authors to include more samples that can ameliorate the statistical analysis and the conclusions. Moreover, including more patients can improve also the correlation between intra-operative cfDNA levels respect to healthy adults.’

Reviewer 1 major revision ii):

‘The experimental approaches conducted in this study include different histotypes in the same analyses. The authors can explain the results also for each histotype in the Results and Discussion section.’

The analysis of STS cohorts containing a variety of histological subtypes is an issue commonly associated with STS research, predominantly due to their rarity. Recognising that STS subtypes should ideally be analysed individually where possible, we have included a subgroup analysis of total cfDNA levels based on this subtype. This is shown in Supplementary figure 2C and commented on in the results section in the sentence below which has been adapted to emphasise this analysis (lines 71-73):

‘No correlation was seen between intra-operative cfDNA concentration and tumour size (R2=0.19), Trojani grade (p=0.9, ANOVA) or STS subtype (p=0.96, ANOVA) (Supplementary Figure 2).’

Reviewer 1 raises an important point with regards to potential variation in ctDNA levels between patients with different STS histotypes.  Overall we thought that reporting the characteristics of individual STS subtypes based on our data alone was unlikely to be particularly informative. This is for 2 reasons – 1) the wide range of STS subtypes included in our modest sized cohort, and 2) our decision to use 3 different experimental approaches to characterise ctDNA. In order to address this point, we have added additional text to our discussion that identifies the STS subtypes that showed strong evidence of ctDNA (i.e. circulating SNVs also detected by comparative WES analysis of tumour tissue) in at least one plasma sample, whilst simultaneously relaying these reservations to the reader.

The new paragraph reads (lines 277-284):

‘We have used three different experimental approaches to detect and monitor ctDNA across a diverse range of STS subtypes, which has reduced our capacity to investigate differences between various subtypes. Nevertheless we note that ctDNA (validated by prior detection of the detected circulating SNVs in matched STS tissue) was detected in one patient with a myxofibrosarcoma (MFS) and two patients with undifferentiated pleomorphic sarcoma (UPS) in our cohort. Circulating tumoural DNA has not been reported previously in patients with an MFS, but it has been detected in up to 30% (2/6) of patients with UPS [19]. Further work will be needed to explore whether patients with UPS are more likely to shed ctDNA than patients with other less common subtypes.’

Reviewer 1 major revisions iii + iv):

‘In the Results section, the Patient-specific ddPCR paragraph appears confused. The somatic SNV identified by comparative WES should be listed (Line 135) to explain the specific SNV tracked in patients.’

‘Table 3 is not included in the manuscript.’

We have liaised with the editing team who have now embedded Table 3 in the revised manuscript. We apologise for the oversight, and believe this table helps to address both of the comments above. Table 3 contains a detailed description of all of the SNVs identified by comparative WES and tracked in matched patient plasma samples using ddPCR assays (including gene name, SNV location, base change, predicted effect, frequency in the tumour). It also tells the reader in which patient(s) each variant was tracked using ddPCR, and whether the variants were successfully detected in their circulation or not.

To further clarify our experimental approach and clear up any reader confusion, we have also made changes to the ‘Patient-specific ddPCR analysis of tumour and ctDNA’ paragraph. The original paragraph below:

‘Given the genetic heterogeneity of STSs and the limited ctDNA detection achieved using our custom designed three-gene ‘Sarcoma V2’ ampliseq panel, we next adopted a personalised approach to detect ctDNA. This involved using patient specific ddPCR assays to track somatic SNVs already identified in patients’ STSs by comparative WES. Serial plasma samples from 8 patients were analysed in this way, with 1 or 2 SNV specific ddPCR assays used in each case (Tables 1 and 3). Despite adopting this new approach, intra- or post-operative ctDNA was only identified in two patients (patient 6 and 22, both diagnosed with UPS).’

Now reads (lines 133-140):

‘Given the genetic heterogeneity of STSs and the limited ctDNA detection achieved using our custom designed three-gene ‘Sarcoma V2’ ampliseq panel, we next adopted a personalised approach to detect ctDNA. For this, patients’ tumours were initially analysed using comparative WES to identify somatic SNVs present. Next, ddPCR assays designed to identify a selection of these variants in the circulation were used to analyse matched patient plasma samples. In total serial plasma samples collected from 8 patients were analysed in this way, with 1 or 2 SNV specific ddPCR assays used in each case (Tables 1 and 3). Despite adopting this new approach, intra- or post-operative ctDNA was only identified in two patients (patient 6 and 22, both diagnosed with UPS).’

Minor revisions

We thank Reviewer 1 for identifying and recommending some minor revisions.

These have all be accepted and changed in the accompanying revised manuscript and supplementary files. The changes have been made using ‘track changes’ so they are identifiable

They include:

In lines 75, 79, 212, in Supplementary Fig 1 and Supplementary Fig 4, the letter P must be lowercase.

  • Done

Line 261, the word UPS should be correct

  • Done (now line 267)

In line 201, “Supplementary 1” should be changed in “supplementary Fig 1”

  • The text in question has been updated to ‘Supplementary Figure 1’ consistent with the rest of the manuscript (line 206).

In line 73, the author should include the statistical correlation.

To address this point, the text below (lines 71-73):

‘No correlation was seen between intra-operative cfDNA concentration and tumour size or Trojani grade and no clear relationship with STS subtype (Supplementary Figure 2).’

has been changed to:

‘No correlation was seen between intra-operative cfDNA concentration and tumour size (R2=0.19), Trojani grade (p=0.9, ANOVA) or STS subtype (p=0.96, ANOVA) (Supplementary Figure 2).’

In line 204, the n. 4 after the refs 22 should be deleted as well as n. 19 in line 281.

  • Done (lines 210 and 295).

In line 146, BRIP1, P971A should be corrected.

  • Now written as BRIP1;P971A consistent with the rest of the manuscript (line 148).

Additional minor changes made by authors:

In addition to the amendments outlined above made in response to the reviewers comments we have also made several other minor changes throughout the manuscript and supplementary files to either correct typos or improve sentence structure. These have been made using track changes so that they can be identified. The most important are listed below:

Main manuscript

Author spelling: Please note the change in the spelling to ‘Aurore Sommer’

Line 205:

‘(range 3.8-43.6ng/ml).’

changed to:

(range 2.3-43.6ng/ml).’

Lines 410-411:

‘The tNGS panels were validated by comparing the frequency of the SNVs derived from WES analysis with the frequency obtained using the AmpliSeq panel’

changed to:

‘The tNGS panels were validated by comparing the frequency of the SNVs derived from WES analysis of patient tumour tissue with their frequency obtained using the AmpliSeq panel.’

Line 448:

‘1. Beckingsale, T.B. ; Shaw, C. (v) Epidemiology of bone & soft tissue sarcomas. Orthopaedics and Trauma 2015, 29, 182-8.’

Changed to:

‘1. Beckingsale, T.B. ; Shaw, C. (v) Epidemiology of bone & soft tissue sarcomas. Orthopaedics and Trauma 2015, 29, 182-8.’

Tables

Table 1 legend:

 ‘DOC-died from other causes;’ removed from legend as the abbreviation is not present in table

Supplementary methods and tables

Evagreen droplet digital PCR:

‘Cycling conditions were 1) 95oC for 5 mins 2) 40 cycles of 30 seconds at 95oC and 60 secs at 95oC 3) 5 mins at 4oC and 4) 5 mins at 90oC.’

corrected to:

‘Cycling conditions were 1) 95oC for 5 mins 2) 40 cycles of 30 seconds at 95oC and 1 min at 60 oC 3) 5 mins at 4oC and 4) 5 mins at 90oC.’

Supplementary Table 5:

‘DOC-died from other causes;’ removed from legend (not present in table)

In summary, we believe the revisions have improved the manuscript and hope that it will now be suitable for publication. Please do not hesitate to contact us if you require any additional information,

Yours Sincerely,

Nick Eastley

Reviewer 2 Report

Overall, the content and quality of the manuscript are excellent. Abstract is sufficient and concise. The introduction is clear and well-organized. The methodology is described appropriately with sufficient details. The sample size is small but that is given due to STSs are rare. The discussion and conclusions are clear. References, tables, and figures are of good quality and readability.

Author Response

We thank reviewer 2 for their very supportive comments.

Additional minor changes made by authors:

In addition to the amendments outlined above made in response to the reviewers comments we have also made several other minor changes throughout the manuscript and supplementary files to either correct typos or improve sentence structure. These have been made using track changes so that they can be identified. The most important are listed below:

Main manuscript

Author spelling: Please note the change in the spelling to ‘Aurore Sommer’

Line 205:

‘(range 3.8-43.6ng/ml).’

changed to:

(range 2.3-43.6ng/ml).’

Lines 410-411:

‘The tNGS panels were validated by comparing the frequency of the SNVs derived from WES analysis with the frequency obtained using the AmpliSeq panel’

changed to:

‘The tNGS panels were validated by comparing the frequency of the SNVs derived from WES analysis of patient tumour tissue with their frequency obtained using the AmpliSeq panel.’

Line 448:

‘1. Beckingsale, T.B. ; Shaw, C. (v) Epidemiology of bone & soft tissue sarcomas. Orthopaedics and Trauma 2015, 29, 182-8.’

Changed to:

‘1. Beckingsale, T.B. ; Shaw, C. (v) Epidemiology of bone & soft tissue sarcomas. Orthopaedics and Trauma 2015, 29, 182-8.’

Tables

Table 1 legend:

 ‘DOC-died from other causes;’ removed from legend as the abbreviation is not present in table

Supplementary methods and tables

Evagreen droplet digital PCR:

‘Cycling conditions were 1) 95oC for 5 mins 2) 40 cycles of 30 seconds at 95oC and 60 secs at 95oC 3) 5 mins at 4oC and 4) 5 mins at 90oC.’

corrected to:

‘Cycling conditions were 1) 95oC for 5 mins 2) 40 cycles of 30 seconds at 95oC and 1 min at 60 oC 3) 5 mins at 4oC and 4) 5 mins at 90oC.’

Supplementary Table 5:

‘DOC-died from other causes;’ removed from legend (not present in table)

In summary, we believe the revisions have improved the manuscript and hope that it will now be suitable for publication. Please do not hesitate to contact us if you require any additional information,

Yours Sincerely,

Nick Eastley

Round 2

Reviewer 1 Report

The authors have addressed properly my suggestions. The manuscript is now suitable for publication.